🔓 | **Open Peer Review** | Human Microbiome | Research Article

# Microbiome gut community structure and functionality are associated with symptom severity in non-responsive celiac disease patients undergoing a gluten-free diet

Laura Judith Marcos-Zambrano,[1] Blanca Lacruz-Pleguezuelos,[1,2] Elena Aguilar-Aguilar,[3,4] Helena Marcos-Pasero,[3,4] Alberto Valdés,[5] Viviana Loria-Kohen,[3,6] Alejandro Cifuentes,[5] Ana Ramirez de Molina,[3] Alberto Diaz-Ruiz,[7,8] Vera Pancaldi,[9] Enrique Carrillo de Santa Pau[1]

**ABSTRACT**  Non-responsive celiac disease (NRCD) challenges clinicians due to persistent symptoms despite a gluten-free diet (GFD). We present a cross-sectional pilot study including 39 NRCD patients to describe the underlying mechanisms contributing to symptom persistence by integrating different levels of data (fecal shotgun metagenomics, mucosal integrity markers, and metabolomic profiles) and using microbial networks to unravel the community structure of the patient's microbiome. Two distinct clusters of patients were identified based on clinical and demographic variables not influenced by gluten consumption. Cluster 1, labeled "Low-grade symptoms," displayed milder symptoms and lower inflammatory markers and a fragmented microbial network characterized by high modularity and a reliance on localized hubs, suggesting a microbial community under stress but capable of maintaining limited functionality. Cluster 2, named "High-grade symptoms," exhibited more severe symptoms, elevated inflammatory markers, and a more connected but antagonistic microbial network with a greater number of keystone taxa, including taxa associated with Th17 activation and inflammation. In contrast, the control network, representing asymptomatic treated celiac disease (tCD) patients, was highly interconnected, resilient, and cooperative, with a robust structure maintained even under simulated disruptions. Metabolomic analysis revealed differential metabolites between clusters, particularly those involved in amino acid metabolism pathways and microbial-derived metabolites such as indolelactic acid and mannitol, which were associated with symptom severity. This study identifies NRCD subgroups based on the gut microbiome and metabolic signatures associated with clinical manifestations, highlighting variations in microbial network stability and metabolite profiles as contributors to symptom persistence and potential therapeutic targets.

**IMPORTANCE** Celiac disease (CD) is a chronic immune-mediated systemic disorder caused by consuming gluten in genetically susceptible individuals. There is currently no cure for CD, and the most effective treatment is maintaining a strict, lifelong gluten-free diet (GFD). This nutritional therapy aims to prevent the immune reaction triggered by gluten and promote the healing of the intestinal lining, resolving the clinical, serological, and histological abnormalities within 6–12 months. However, up to 30% of patients may continue to experience symptoms or exhibit laboratory abnormalities or intestinal inflammation suggestive of active CD, despite following a GFD. This challenge, which encompasses various diagnoses, is known as nonresponsive celiac disease (NRCD). In this study, we explored the role of intestinal microbiota in causing NRCD, finding an association between the persistence of symptoms and changes in mucosal integrity

**Peer Reviewers** Yan Ni, The Children's Hospital, HangZhou, China; Jing Wu, Nanjing University of Chinese Medicine, Nanjing, China

Address correspondence to Laura Judith Marcos-Zambrano, judith.marcos@imdea.org.

The authors declare no conflict of interest.

See the funding table on p. 22.

biomarkers, with different gut microbiome structures among NRCD patients, indicating a significant role of the microbiome in NRCD.

**KEYWORDS**    microbiome, metabolome, celiac disease, gluten-free diet, co-occurrence networks, symptom association

Celiac disease (CD) is a multifaceted autoimmune disorder that arises as a response to the consumption of gluten, the primary protein stored in wheat, barley, and rye (1). This condition primarily affects individuals with a genetic predisposition and leads to increased levels of autoantibodies specific to CD. Consequently, it manifests as varying degrees of inflammation in the small intestine and gives rise to a broad spectrum of gastrointestinal and extra-intestinal symptoms (2).

Currently, the only effective approach for managing CD involves strict and lifelong compliance with a gluten-free diet (GFD) (3). Typically, this course of action leads to the alleviation of inflammation in the small intestine (1). Despite a significant improvement being observed in most individuals with CD shortly after eliminating gluten from their diet, a notable percentage of individuals ranging from 7% to 30% continue experiencing symptoms or exhibiting clinical indications suggestive of CD, despite adhering to a GFD (4). This clinical challenge, encompassing a diverse array of diagnoses, is referred to as nonresponsive celiac disease (NRCD) (5).

NRCD can be further categorized as primary if there is an initial lack of response to a GFD or secondary if signs, symptoms, or laboratory abnormalities consistent with CD reoccur following an initial period of normalization while maintaining a GFD (6). In previous studies, the primary contributor to NRCD was unintentional ingestion of gluten, accounting for approximately 50% of cases (6, 7). However, the etiologies of NRCD exhibit significant variability and can include lymphoma, small-intestinal bacterial overgrowth (SIBO), microscopic colitis, pancreatic insufficiency, disaccharidase deficiency, and irritable bowel syndrome (IBS) (6, 8, 9). NRCD has been poorly studied; a clear cause–effect association to explain this entity is still lacking.

Recently, it has been observed that gut microbiota differs among subgroups of CD patients according to specific clinical manifestations and symptoms (10), which suggests that gut microbiota plays an essential role in the persistence of symptoms in nonresponsive patients after adherence to a GFD. It is well known that patients with CD show dysbiosis, with a decrease in species from genus *Bifidobacterium* and an overrepresentation of proinflammatory groups, a picture that is likely to precede the onset of the disease and persist after diagnosis, at least in some cases (11). Considering that diet is one of the major drivers of human gut microbiota composition (12), it is reasonable to think that GFD can directly affect the host's physiology and metabolism, as well as gut bacterial activity. For this reason, the interactions between the host's genetics, GFD, and microbiota need to be studied in depth. Notably, nutritional regimens, including GFD, involve the production of specific microbial and human metabolites (13).

Here, we investigate the microbial community structure among NRCD patients through the application of systems biology methodologies, particularly employing microbial co-occurrence networks to visualize and comprehend the relationships and coexistence patterns among various microorganisms in the intestinal microbial ecosystem. Additionally, we conducted an exhaustive analysis of the intestinal metabolome in these individuals. Our main goal is to integrate the identified microbial and metabolomic alterations with the persistent manifestation of symptoms.

## MATERIALS AND METHODS

We followed the reporting guidelines proposed by the "Strengthening The Organization and Reporting of Microbiome Studies" (STORMS) consortium (14). The STORMS checklist is provided in Data S1.

## Study subjects from IMDEA Food Institute

The cross-sectional pilot study was performed at the Nutrition and Clinical Trials Unit, GENYAL Platform in the IMDEA Food institute (Madrid, Spain) in 2017. A purposive sampling strategy was used to engage participants. Inclusion criteria were men and women aged between 18 and 65 years, those who agree to participate in the study voluntarily and give written informed consent, those who have CD confirmed by intestinal biopsy or documented medical diagnosis at least 12 months before the study, and are following a GFD for at least 12 months. The inclusion criteria also include the presence of any of the following symptoms: diarrhea, soft stools or constipation, abdominal pain, bloating, nausea, discomfort from noise and bowel movements, and tenesmus. Exclusion criteria were individuals with other pathologies of the digestive system (Crohn's disease, irritable bowel syndrome [IBS], ulcerative colitis, and colon cancer); individuals who have undergone digestive system surgery (e.g., short bowel syndrome); subjects with serious diseases (liver, kidney, and cancer) or autoimmune diseases or who use immunosuppressive or anti-inflammatory drugs; subjects with dementia, mental illness, or decreased cognitive function; subjects usually treated with probiotics or taking antibiotics; pregnant or breastfeeding women; or those with any other condition that limits compliance with procedures established in the protocol, as well as adherence to treatment by the subject. A total of 39 patients were enrolled in the study.

## Ethical aspects and data processing

Protocols and methodology used in the present study comply with the ethical principles for research involving human subjects laid down in the Declaration of Helsinki (1964) and its modifications. The study was approved by the Research Ethics Committee of the IMDEA Food Foundation (PI-032 approval date: 12 June 2017). Participants were informed in detail about the different stages of the project both orally and in writing. The researchers collected signed informed consent prior to the first evaluation. This document included a specific consent to microbial profiling. Data compiled along the study were processed by applying dissociation criteria, making the volunteers' data anonymous, in compliance with the current Spanish and European legislation (Regulation (EU) 2016/679 and Organic Law on Personal Data Protection and Guarantee of Digital Rights 3/2018).

## Symptomatology and quality of life assessment

Symptomatology data were recorded through validated questionnaires. (i) The Gastrointestinal Symptom Rating Scale Celiac Disease (GSRS) (15) is a validated 15-item questionnaire employed to assess the severity of gastrointestinal symptoms associated with CD and other related gastrointestinal conditions. The survey comprises five distinct sub-dimensions: indigestion, diarrhea, abdominal pain, reflux, and constipation. Scoring for each sub-dimension involves calculating the mean value of the respective items, while the overall GSRS-CD score is determined by computing the mean value across all 15 items. The scoring is performed on an 11-point Likert scale, ranging from 0 to 10, where higher scores indicate more pronounced symptom severity. (ii) Celiac Disease Patient-Reported Outcome (CeD-Pro) (16) is a validated 10-item questionnaire used to evaluate the severity of gastrointestinal and extraintestinal symptoms from CD. Responses are scored on an 11-point (0–10) Likert scale, with higher scores indicating greater symptom severity. Quality of life was recorded through the Celiac Disease Quality of Life (CD-QOL) survey (17), a validated 20-item questionnaire with four factor analytically derived subscales: limitations, dysphoria, health concerns, and inadequate treatment scored with a Likert scale from 0 to 5.

## Dietary intake and gluten-free diet compliance

A validated food record of 72 hour consumption (18) in which participants had to sum up all food and drinks ingested during three days (2 weekdays and 1 Sunday or holiday) was delivered. Afterward, the data were tabulated and analyzed by a certified dietician using the DIAL nutritional software v 3.15 (Alce Ingeniería, Madrid, Spain) in order to obtain information about macro and micronutrients and calculate the healthy eating index (19). Gluten-free diet compliance was monitored by the celiac dietary adherence test (CDAT) (20), a validated questionnaire that considers five crucial dimensions pertaining to compliance with a GFD: the presence of CD symptoms, the patient's understanding of the disease and its treatment, confidence in the effectiveness of the treatment, motivating factors for adhering to a GFD, and self-reported adherence to the diet. The questionnaire comprises seven items, each rated on a 5-point Likert scale, resulting in a total score ranging from 7 to 35 points. The scoring interpretation is as follows: a score of 7 points indicates excellent GFD adherence; 8–12 points suggest very good GFD adherence; 13–17 points imply insufficient or inadequate GFD adherence, and scores exceeding 17 points indicate poor GFD adherence.

## Anthropometric and laboratory measurements

Anthropometric measurements were determined early in the morning, by previously trained dieticians, following standardized protocols. Height was determined using a Leicester height rod with millimetric accuracy (Biological Medical Technology SL, Barcelona, Spain). Body weight, fat mass (FM) percentage, and muscle mass (MM) percentage were assessed using a body composition monitor (BF511; Omron Healthcare Co., Ltd., Kyoto, Japan). Waist circumferences (WC) were taken using a nonelastic tape (KaWe Kirchner & Wilhelm GmbH, Asperg, Germany; range 0–150 cm, 1 mm of precision). Measurements were taken twice in a row, considering the average as the result. For blood pressure monitoring, an automatic digital monitor was used (OMRON M3-Intelli-sense).

Venous blood samples were collected at the baseline in evacuated plastic tubes (VACUETTE TUBE; Greiner Bio‑One GmbH, Kremsmünster, Austria). Total cholesterol (TC), low-density lipoprotein cholesterol (LDL‑C), high‑density lipoprotein cholesterol (HDL-C), triglycerides, apolipoprotein A1, apolipoprotein B (APOB), hs‑CRP, HbA1c, inflammatory markers (IFN-$\gamma$, TNF-a, IL-10, IL-12, IL-1, IL-6, and IL-15), and markers associated with mucosal integrity (I-FAB, calprotectin, lactoferrin and citrulline, lactulose, mannitol, and D-lactate) were measured according to a standardized protocol by Laboratory CQS Consulting (UNE-ISO 15189:2007 (No. 659/LE1318).

## Microbial profiling

Fresh feces samples were collected in a collection tube with no preservatives. Samples were directly brought by the participants under cold conditions (ice packs) within 2–4 hours of collection. Upon receipt, the feces were stored at −80°C for analysis. DNA was extracted using the QIAmp DNA stool Minikit according to the manufacturer's instructions (QIAGEN, Hilden, Germany). Microbial analysis of the samples was performed by whole-genome shotgun sequencing on the NextSeq Illumina platform, by using a Mid Output kit 2 × 150 pb, and a coverage of ~24 million reads per sample.

The metagenomic analysis was performed following the general guidelines and relying on the bioBakery computational environment after filtering form host-DNA with BowTie2 version 2.3.4.3. The taxonomic profiling and quantification of organisms' relative abundances of all metagenomic samples have been quantified using MetaPhlAn 3.062. Metagenomes were mapped internally in MetaPhlAn 3.0 against the marker genes database with BowTie2 version 2.3.4.3 with the parameter "very-sensitive." The resulting alignments were filtered to remove reads aligned with an MAPQ value <5, representing an estimated probability of the likelihood of the alignments. Functional potential analysis of the metagenomic samples was performed using HUMAnN2 (version 0.11.2 and UniRef

database release 2014-07) that computed pathway profiles and gene–family abundances.

Compositional data transformation was used for further analysis. To estimate the microbiome species richness of an individual from the taxonomic profiles, we computed three alpha diversity measures: the number of species found in the microbiome ("observed richness"), the Shannon entropy estimation, and the Simpson dominance index using phyloseq (21). Microbial beta diversity was calculated with the weighted Unifrac distance (22), and differences between groups were assessed with the PERMANOVA test using the adonis function from the vegan R package (version 2.6-6). For analyzing the identified metabolic pathways in the microbiome, we used the statistical analysis of taxonomic and functional profiles (STAMP) software (23) which uses statistical hypothesis tests: Welch's $t$-test and White's nonparametric $t$-test for comparing profiles organized into two groups.

## Study controls from public data

To include controls, we searched for publicly available data from patients with CD confirmed by intestinal biopsy or documented medical diagnosis. These patients were required to have been following a strict GFD for at least 12 months, to be asymptomatic (as determined through clinical history or validated clinical questionnaires), and to have negative results for anti-tissue transglutaminase IgA (anti-tTG-IgA) or anti-tTG/deamidated gliadin peptide IgG (anti-tTG/ADGP-IgG) in cases of IgA deficiency.

A bibliographic search was performed using PubMed and SCOPUS databases with the following search terms: (((celiac disease) AND (metagenomics)) NOT (mice)) NOT (review[Publication Type]). We also explored data sets available in the MGNify (24) and curatedMetagenomicData (25) repositories. Figure S1 presents the PRISMA flowchart summarizing the selection process. The initial search identified 44 studies of interest. After removing duplicates, studies related to Crohn's disease, a review article, and a study conducted in primates, 29 studies remained for further screening. Of these, only three met the eligibility criteria: conducted in adults, involving fecal samples, and analyzing the microbiome through whole-genome sequencing data. However, only one study included the necessary metadata to classify patients as asymptomatic controls and was ultimately selected for our analysis.

The selected study by Francavilla et al. (26) included patients from the Gastroenterology Unit of Ospedale Mauriziano Umberto I and the Gastroenterology outpatient clinic of San Giovanni Antica Sede, in Turin, Italy. Data from asymptomatic treated celiac disease patients (tCD) were included as controls. Specifically, we selected 48 individuals who adhered strictly to a GFD, tested negative for transglutaminase (TG) serology (tCD-TG-), and met all inclusion criteria. Initially, metagenomic data were available for 50 tCD-TG- samples. However, data from two samples were unrecoverable due to file corruption, preventing their inclusion in the analysis. Consequently, the final data set comprised 48 tCD-TG samples. Metagenomic samples from this study were downloaded from the Sequence Read Archive (SRA) under the identifier PRJNA904924. These samples were processed using the same pipeline as the data collected in the present study, as previously described. Metadata were obtained from supplemental materials associated with the selected study.

## Batch effect correction

Microbial species with an abundance above 0.01% in 10% of the samples were kept for further analyses. The MMUPHin R package (version 1.14.0) was used for batch effect correction caused by the study of origin while controlling for the effect of the disease–control group. The effect of batch adjustment was evaluated based on the total variability in microbial profiles attributable to differences in the study of origin. This was done with a PERMANOVA test using the adonis function from the vegan R package (version 2.6-6).

## Microbial co-occurrence networks

Microbial co-occurrence networks were constructed using the sparse inverse covariance estimation for ecological association and statistical inference (SpiecEasi) method (27) with the NetCoMi R package (28). Nodes represented the microbial species, and edges reflected their co-occurrence across individuals, as measured by the correlation of their abundance across patients.

### Network visualization and analysis

Networks were visualized using the "spring" layout in NetCoMi, which employs the Fruchterman–Reingold algorithm (29) to create a force-directed layout. Analyses of the networks were performed with the NetworkX Python library (version 3.4.2). The largest connected component (LCC) of each network was analyzed to assess the topology and stability. Network topology was evaluated using commonly used metrics such as degree (number of connections), node betweenness centrality, average shortest path length, and the number of connected components, indicating the number of distinct subnetworks produced by SpiecEasi for each phyloseq object. The average shortest path length measured the mean distance between all pairs of nodes in a network, while node betweenness centrality quantified the fraction of shortest paths passing through a given node. Keystone taxa were defined as highly connected taxa that significantly influenced microbiome structure and function, regardless of their abundance (30). These were determined as hubs (nodes with high degree centrality) and bottlenecks (nodes with high betweenness centrality) with a degree and betweenness centrality greater than the log-normal 90 quantiles.

### Network stability

Network stability analysis was conducted in Python using NetworkX. Negative edges were excluded, focusing solely on positive microbe–microbe interactions. Nodes were ranked by the properties of interest (e.g., degree and betweenness centrality), and the highest-ranked node was sequentially removed. After each removal, the size of the LCC was recalculated. This process continued until all nodes were disconnected. For random attacks, nodes were selected randomly using the choice method from Python's random module, repeating the process 1,000 times to ensure robustness of the results.

### Network comparison

Nonparametric standard permutation tests were used to assess differences between centrality measures and global network characteristics between each group. A differential association network was constructed using Pearson correlation and Fisher $Z$ test, using FDR-adjusted $P$-values to only include the differentially associated species between clusters.

## Sample preparation for metabolomics and UHPLC-Q/TOF-MS/MS procedure

Metabolomics data were only available for subjects from the IMDEA Food Institute. Stool samples were stored in aliquots at −80°C and thawed on ice prior to analysis. The weighted and dry samples were mixed with methanol 80% (1:10, wt/vol) and mixed for 5 minutes, and then ultrasound was applied for 30 minutes at room temperature. Finally, samples were centrifuged at 14,800 rpm for 30 minutes at 4°C, and the supernatant was collected and divided into two fractions for their analysis by UHPLC-Q/TOF-MS/MS combining two analytical methods.

### UHPLC-C18-Q/TOF-MS/MS

An aliquot of 2 µL from the first fraction was injected into a HPLC model 1290 (Agilent Technologies, Germany), and compounds were separated using an Agilent Zorbax Eclipse Plus C18 column (100 mm length × 2.1 mm id; 1.8 µm particle size) equipped with

an Agilent Zorbax C18 pre-column (5 mm × 2.1 mm id, 1.8 µm). The column temperature was held at 40°C, and the flow rate was set at 0.5 mL/minute. Milli-Q water was used as mobile phase (A) and acetonitrile (ACN) as mobile phase (B), and 0.1% formic acid was used as a mobile phase modifier. The gradient elution program was set up as follows: 0% to 30% B in 7 minutes; 30% to 80% B in 2 minutes; 80% to 100% B in 2 minutes; 100% B for 2 minutes; 100% to 0% B in 1 minutes; 0% B for 3 minutes. Compounds were eluted into a Q/TOF series 6540 from Agilent Technologies (Germany), equipped with an Agilent Jet Stream thermal orthogonal ESI source. The mass spectrometer was operated in the ESI positive mode using the following parameters: capillary voltage, 3,000 V; mass range, 25–1,100 $m/z$; nebulizer pressure, 40 psig; drying gas flow rate, 8 L/minute; dry gas temperature, 300°C. The sheath gas flow was 11 L/minute at 350°C. Method blanks and a pooled mixture of all samples were included as quality control samples and were subjected to iterative MS/MS with a mass error tolerance of 20 ppm and retention time (RT) exclusion tolerance of ±0.2 minutes to increase the coverage of the MS/MS spectra acquired. MS/MS spectra were acquired employing the auto MS/MS mode using five precursors per cycle, dynamic exclusion after two spectra (released after 0.5 min), and collision energies of 20 V and 40 V. Mass accuracy was corrected using ions at $m/z$ 121.0509 ($C_5H_4N_4$) and 922.0098 ($C_{18}H_{18}O_6N_3P_3F_{24}$) simultaneously pumped into the ion source.

### UHPLC-HILIC-Q/TOF-MS/MS

Fifty microliters of the second fraction was evaporated and resuspended in 40 µL of ACN:water (80:20, vol/vol) with a mixture of internal standard compounds (CUDA, DL-alanine-d3, DL-glutamic acid-d3, d9-choline chloride, 15N2-L-arginine, L-methionine-d8, and Val-Tyr-Val). An aliquot of 5 µL was injected in the same instrument as specified above, and compounds were separated using a Waters Acquity UPLC BEH Amide column (150 mm × 2.1 mm id; 1.7 µm particle size) equipped with a Waters Acquity UPLC BEH Amide VanGuard Pre-column (5 mm × 2.1 mm id, 1.7 µm). The column temperature was held at 45°C, and the flow rate was set at 0.4 mL/minute. Milli-Q water was used as mobile phase (A), and 95:5 (vol/vol) ACN:water was used as mobile phase (B), and 10 mM ammonium formate and 0.125% formic acid were used as mobile phase modifiers. The gradient elution program was set up as follows: 100% B for 2 minute; 100% to 70% B in 5.7 minutes; 70% to 40% B in 1.8 minutes; 40 to 30% B in 0.75 minutes; 30% to 100% B in 2.5 minutes; 100% B for 4 minutes. Compounds were eluted into the same mass spectrometer, as described before. The mass spectrometer was operated in the ESI negative mode using the following parameters: capillary voltage, −3,500 V; mass range, 50–1,700 $m/z$; nebulizer pressure, 35 psig; drying gas flow rate, 11 L/minute; dry gas temperature, 200°C. The sheath gas flow was 11 L/minute at 350°C. Method blanks and a pooled mixture of all samples were included as quality control samples and were subjected to iterative MS/MS using the same parameters as described before. Mass accuracy was corrected using ions at $m/z$ 119.0363 ($C_5H_4N_4$) and 966.0007 ($C_{18}H_{18}O_6N_3P_3F_{24}$) simultaneously pumped into the ion source.

### Metabolomics data processing

Data obtained from each analytical platform (UHPLC-C18-Q/TOF-MS/MS and UHPLC-HILIC-Q/TOF-MS/MS) were converted to ABF format and processed separately by MS-DIAL software (v. 4.8). In-house RT-$m/z$ libraries, the public LipidBlast MS/MS spectral library, and the NIST20 MS/MS database were used for metabolite annotation. For UHPLC-C18-Q/TOF-MS/MS, the following parameters were used: retention time, 0–14 minutes; mass range, 25–1,100 Da; MS1 tolerance, 0.01 Da; MS1 tolerance, 0.025 Da; minimum peak height, 1,000; accurate mass tolerance (MS1 and MS2) for MSP library, 0.01 Da and 0.025 Da; identification score cutoff for MSP library, 80%; retention time tolerance for alignment, 0.1 minute; MS1 tolerance for alignment, 0.015 Da. The parameters used for processing UHPLC-HILIC-Q/TOF-MS/MS data were as follows: retention time, 0–17 minutes; mass range, 50–1,700 Da; MS1 tolerance, 0.01 Da; MS1

tolerance, 0.025 Da; minimum peak height, 1,000; accurate mass tolerance (MS1 and MS2) for MSP library, 0.01 Da and 0.025 Da; identification score cutoff for the MSP library, 80%; retention time tolerance for RT-$m/z$ library, 0.1 minute; accurate mass tolerance for RT-$m/z$ library, 0.01 Da; identification score cutoff for RT-$m/z$ library, 85%. CUDA, DL-alanine-d3, DL-glutamic acid-d3, choline-d9, $^{15}N_2$-L-arginine, L-methionine-d8, and Val-Tyr-Val internal standard compounds were used for retention time correction. Metabolites were annotated following the Metabolomics Standard Initiative (MSI) guidelines (31): MSI level 1 for metabolites with precursor $m/z$, in-house RT-$m/z$ libraries, and MS/MS spectral library matching; MSI level 2 a for metabolites with precursor $m/z$ and MS/MS spectral library matching, and MSI level 2b for metabolites with precursor $m/z$ and in-house RT-$m/z$ library matching. Unknown metabolites, metabolites with a maximum peak height below three times the average height in the blank samples, and metabolites with a maximum height below 1,000 units were removed. Finally, metabolite height normalization from the two analytical methods was performed by using the systematical error removal using random forest (SERFF) method (https://slfan2013.github.io/SERRF-online/) and using the signals from the pooled mixtures.

## Host–microbiome metabolome analysis

Host–microbiome analysis was performed using the online tool MetOrigin (32), which integrates gut microbiome data and identifies host, microbiome, and co-metabolism activities associated with microbial metabolites. Microbial species from shotgun metagenomics and metabolomics data were filtered based on a low RSD threshold (30%) and normalized using bacterial relative abundance (percentage) and log-transformed metabolites. The KEGG database was used for the human metabolic pathway curation, with $P$-value calculated based on a hypergeometric test. Metabolic pathways with log2 $P$-values > 1 were considered statistically significant and subjected to correlation analysis (Spearman correlation method, $P$-value < 0.05). MetOrigin employed Sankey network diagrams to demonstrate associations between microorganisms and metabolites. The analysis generated microbial and metabolite interaction networks by integrating differential metabolites from the host, microbiota, and co-metabolic sources, along with their related bacteria.

## Statistics

All statistical analyses were performed using R version 4.1 (33).

### Multiple-factor analysis

MFA was performed using FactoMineR (34) and factoextra R (35) packages, over the symptomatology, dietary intake, GFD compliance, anthropometric, and metabolic measurements to select the most informative variables. Afterward, a Gaussian mixture model for clustering over the Canberra distance of the selected variables was performed by using the mclust R package (36).

### Between-group comparisons

Independent two-sample $t$-tests or Wilcoxon rank-sum exact tests (in cases where residuals were not reasonably normally distributed) were performed to compare variables between patient groups.

### Correlation analysis

Pearson's correlation was run to assess associations between changes in gastrointestinal symptoms and changes in the gut microbiome (keystone and differentially associated taxa), microbial functional potential (differential abundance metaCyc pathways and GO modules), and fecal metabolome (differential abundant metabolites).

**TABLE 1** Baseline characteristics from 39 patients with NRCD included in the study[a]

| Characteristic | N (%) or mean (range) |
| --- | --- |
| Male | 5 (13%) |
| Female | 34 (87%) |
| Age (years) | 35 (18–60) |
| Weight (kg) | 61 (44–91.5) |
| Height (cm) | 164 (152.5–181) |
| BMI (m/kg$^2$) | 22.6 (16.7–34) |
| Years since diagnosis | 10 (1–27) |
| CDAT score | 12 (7–18) |
| Years under GFD | 10 (1–28) |
| Positive biopsy | 34 (87%) |
| Positive antibodies | 32 (82%) |
| Positive HLA study | 17 (45%) |
| CD-QOL score | 71 (46–93) |
| Number of different symptoms | |
| 1 | 6 (15%) |
| 2 | 9 (23%) |
| 3 | 6 (15%) |
| 4 | 5 (14%) |
| 5 | 7 (18%) |
| >6 | 6 (15%) |

[a]BMI, body mass index. CDAT, Celiac Dietary Adherence Test. GFD, gluten-free diet. HLA, human leukocyte antigens. CD-QOL, Celiac Disease Quality of Life.

### Statistical significance

For statistical testing, adjusted $P$ values below 0.05 were considered statistically significant.

## RESULTS

The baseline characteristics of the 39 participants in the study are displayed in Table 1. Briefly, most participants were female (87%), with an average age of approximately 35 years. All participants exhibited a BMI indicating normal weight. Furthermore, all participants have a confirmed medical diagnosis of celiac disease by biopsy (87%), antibodies (82%), and/or HLA study (45%). The mean time since diagnosis was 7.1 years, and the mean time for undergoing a GFD was 7.2 years (range 1–28 years). Moreover, all participants reported experiencing gastrointestinal or extraintestinal symptoms characteristic of CD, including cramps, diarrhea, nausea, flatulence, headache, and fatigue, with an average symptom count of three different symptoms (range 1–8 different symptoms).

Gluten consumption was measured by the Celiac Dietary Adherence Test (CDAT). A mean score of 12 was obtained (range 7–18), indicating very good GFD adherence. Overall fecal gluten immunogenic peptides (GIP) levels were negative (mean value 0.123 ng/g), indicating no gluten consumption among the study participants.

### NRCD subgroups can be defined according to clinical and demographic variables not influenced by gluten consumption

Multiple factorial analysis was used to determine the importance of the clinical, demographic, inflammatory, and mucosal integrity biomarkers. From the initial set of 99 variables analysed in the study, 47 were chosen based on their informativeness (Data S2). These selected variables were subjected to a Gaussian mixture model analysis, utilizing the Canberra distance metric, leading to the identification of two distinct clusters.

Cluster 1, designated as "Low-grade symptoms" (low-NRCD), exhibited lower levels of symptoms and inflammatory markers. Conversely, Cluster 2, termed "High-grade

**TABLE 2** Barrier integrity, inflammatory markers, and symptoms from patients with low-grade symptomatology (low-NRCD) and NRCD with high-grade symptomatology (high-NRCD)

| Characteristic[a] | Low-NRCD (n = 25)[b] | High-NRCD (n = 14)[b] | P-value[c] |
|---|---|---|---|
| I-FABP (pg/mL) | 292 (140) | 221 (187) | 0.5 |
| GIP (ng/g) | 0.25 (0.13) | 0.10 (0.02) | 0.16 |
| Citrulline (µmol/L) | 1.57 (0.54) | 1.03 (0.03) | 0.4 |
| Lactulose:mannitol | 0.78 (0.94) | 1.63 (1.35) | 0.8 |
| Fecal calprotecnin (µg/g) | 25 (12) | 31 (24) | 0.11 |
| Anti-GP2 IgA | 1.90 (0.67) | 1.70 (0.52) | 0.5 |
| IFN (pg/mL) | 21 (11) | 27 (12) | 0.2 |
| IL-1 (pg/mL) | 1.18 (0.59) | 1.64 (0.74) | 0.068 |
| IL-10 (pg/mL) | 6.6 (4.4) | 10 (5.2) | 0.071 |
| IL-12 (pg/mL) | 2.37 (1.06) | 3.57 (1.19) | **0.016** |
| IL-6 (pg/mL) | 1.10 (0.82) | 1.62 (0.56) | 0.10 |
| IL-15 (pg/mL) | 20 (17) | 15 (11) | 0.3 |
| TNFa (pg/mL) | 2.98 (1.13) | 4.04 (1.18) | **0.006** |
| Number of symptoms | 2.68 (1.46) | 4.85 (1.66) | **<0.001** |
| Cramps | 2.07 (2) | 5.64 (4.88) | **0.001** |
| Abdominal pain | 2.73 (2) | 5.79 (2.88) | **<0.001** |
| Abdominal swelling | 3.68 (2) | 7.07 (2.38) | **<0.001** |
| Constipation | 3.27 (5.5) | 2.79 (2.62) | 0.6 |
| Diarrhea | 0.92 (1.5) | 5.5 (5.88) | **<0.001** |
| Flatulence | 3.08 (1.5) | 7.64 (1.88) | **<0.001** |
| Loose stools | 1.29 (2) | 5.75 (4.25) | **<0.001** |
| Headache | 2.84 (4) | 5.68 (5) | **0.010** |
| Fatigue | 3.66 (2) | 7.11 (1.88) | **<0.001** |

[a]I-FABP: serum intestinal fatty acid binding protein. Anti-GP2 IgA: anti-glycoprotein 2IgA. IFN: interferon, IL: interleukin, TNFa: tumor necrosis factor alpha.
[b]Mean (standard deviation).
[c]Wilcoxon rank-sum test. Boldface indicates significant corrected P-values.

symptoms" (high-NRCD), displayed elevated scores in patient-reported symptom questionnaires, heightened inflammatory markers, and increased intestinal permeability. A comprehensive summary of these findings can be found in Table 2.

An in-depth analysis of the symptomatology revealed noteworthy distinctions between the two clusters of NRCD patients. Patients in the low-NRCD group experienced an average of two different symptoms, while patients in the high-NRCD group reported up to five different symptoms ($P < 0.01$). Particularly, individuals assigned to the high-NRCD group displayed significantly more severe gastrointestinal symptoms, including cramps, abdominal pain, abdominal swelling, constipation, diarrhea, flatulence, and loose stools, as assessed by two distinct questionnaires ($P < 0.001$). Furthermore, patients in the high-NRCD group reported a higher incidence of extraintestinal symptoms such as headaches and fatigue ($P < 0.001$).

To further investigate the integrity of the gastrointestinal barrier in terms of inflammation, functionality, and enterocyte permeability, several measurements were conducted. Initially, we assessed levels of intestinal fatty acid binding protein (I-FAB) and intestinal calprotectin, both markers associated with inflammation and damage to intestinal cells. We found that both parameters were within normal limits for both NRCD patients' groups. I-FAB levels were 292 vs 221 pg/mL ($P$ = ns; normal range, <1 ng/mL), while fecal calprotectin levels were 25 vs 31 µg/g ($P$ = ns; normal range, <50 µg/mg). Urinary citrulline levels were close to the lower limit in both groups (1.57 vs 1.03 µmol/L, $P$ = ns; reference value, 1–46 µmol/L). Low levels of citrulline are associated with dysfunction of the small intestine and advanced degrees of villous atrophy. However, the most relevant value was the lactulose:mannitol ratio (0.78 vs 1.63, $P$ = ns; reference value, <0.03), representing an approximately 20-fold increase compared to values observed in healthy controls (37). Elevated values in this ratio suggest an increase

**TABLE 3** Dietary data of patients with low-grade symptomatology (low-NRCD) and NRCD with high-grade symptomatology (high-NRCD)

| Characteristic[a] | Low-NRCD (n = 25)[b] | High-NRCD (n = 14)[b] | P-value[c] |
|---|---|---|---|
| CDAT | 12 (3) | 13 (3) | 0.12 |
| HEI | 55 (10) | 53 (13) | 0.4 |
| Diet variety | 8 (2.8) | 7.93 (3.08) | 0.8 |
| TEI (kcal/day) | 2,094 (532) | 2,201 (519) | 0.6 |
| Proteins (g/day) | 82 (20) | 78 (21) | 0.5 |
| Carbohydrates (g/day) | 184 (64) | 194 (49) | 0.7 |
| Lipids (% TEI) | 45 (7) | 47 (11) | 0.4 |
| Proteins (% TEI) | 16.03 (3.06) | 14.36 (2.72) | **0.044** |
| Solube fiber (g/day) | 2.72 (1.34) | 3.29 (2.30) | 0.6 |
| Insoluble fiber (g/day) | 5.57 (2.33) | 6.53 (3.77) | 0.4 |
| Vegetal fiber (g/day) | 22 (6) | 22 (10) | 0.7 |
| Zinc (mg/day) | 8.04 (2.68) | 7.84 (2.03) | >0.9 |
| Calcium (mg/day) | 717 (268) | 678 (230) | 0.7 |
| Iron (mg/day) | 10.7 (3.5) | 9.7 (3.3) | 0.7 |
| Magnesium (mg/day) | 237 (64) | 247 (90) | 0.9 |
| Vitamin K (µg/day) | 144 (84) | 126 (70) | 0.7 |
| Vitamin D (µg/day) | 3.7 (3.7) | 2 (2.7) | **0.022** |
| Vitamin B12 (µg/day) | 6.89 (10.08) | 4.16 (2.236) | 0.3 |
| Vitamin A (µg/day) | 1,305 (2,506) | 815 (452) | 0.7 |
| Vitamin E (µg/day) | 10.3 (4.1) | 13.8 (8.7) | 0.4 |
| Folic acid (µg/day) | 226 (93) | 203 (94) | 0.6 |

[a]CDAT: Celiac Dietary Adherence Test. HEI: healthy eating index. Diet variety, number of different food items. TEI: total energy intake of total caloric value of the diet.
[b]Mean (standard deviation).
[c]Wilcoxon rank-sum test. Boldface indicates significant corrected P-values.

in the intestinal permeability, which may lead to the passage of molecules associated with an exacerbated inflammatory response into the bloodstream.

In terms of inflammatory biomarkers, elevated levels of the proinflammatory cytokines IL-15 (20 vs 15 pg/mL, $P$ = ns) and IFNγ (21 vs 27 pg/mL, $P$ = ns) were observed in patients from both NRCD groups. These cytokines are known to play a crucial role in the development of celiac disease by promoting immune activation and causing damage to the intestinal mucosa. Additionally, we found higher levels of the proinflammatory cytokines IL-12 (2.37 vs 3.57 pg/mL, $P < 0.01$) and TNFa (2.97 vs 4.04 pg/mL, $P < 0.01$) in patients in the high-NRCD group. These findings indicate a potential association between the high-NRCD group and heightened inflammatory activity.

## Diet quality was similar in both patient clusters

The adoption of GFD may induce a potential imbalance in nutrient intake, thereby compromising the nutritional adequacy of the diet, primarily due to the exclusion of various fiber-rich food sources. Assessment of the micro and macronutrient consumption of participants was conducted through the administration of self-reported questionnaires under the supervision of a qualified dietician.

Both groups demonstrate adequate intake percentages (above 80%) for major vitamins. However, it is noteworthy that vitamin D consumption was low (74% and 39% adequacy relative to the dietary reference values (DRV) for each cluster; $P < 0.05$). The adequacy percentage for zinc, iron, and calcium was around 60% of the DRV, with no significant differences.

The diet quality of patients in both NRCD groups was fair, as evidenced by a healthy eating index (HEI) close to 50% (low-NRCD: 55% vs high-NRCD: 53%; $P$ = ns). Deficiencies primarily arise from low cereal consumption and limited dietary variety (Table 3). Regarding macronutrient intake, both groups display similar consumption levels of fats, carbohydrates, and simple sugars. However, the percentage of protein relative to

the total energy intake of the diet was lower in patients from the high-NRCD group compared to the low-NRCD group.

## Microbial community structure differed between the two clusters

We included publicly available data from a study involving 48 asymptomatic treated celiac disease patients (tCD) as controls to investigate the structure and diversity of the gut microbial community. After correcting for batch effects (Fig. S2), we analyzed all data sets together, identifying 431 microbial species. The relative abundance analysis revealed that the microbial community was dominated by the phyla *Bacteroidetes*, *Firmicutes*, *Verrucomicrobia*, *Proteobacteria*, and *Actinobacteria*. The microbial community composition was largely similar across the three groups, although small differences were identified in specific markers following a linear discriminant analysis (Fig. 1C). Regarding alpha diversity indices, we observed higher diversity (Chao1) in the tCD group compared to the NRCD groups. However, the Shannon and Simpson indices were comparable across all three groups, indicating similar evenness in the microbial distribution (Fig. 1A). We calculated the beta diversity using the Unifrac distance to observe the differences between the three study groups (Fig. 1B). We found that although the differences in microbial composition between the groups were statistically significant ($P < 0.05$), the effect of the groups was small since the variation explained by them is limited (R = 0.034), suggesting that the groups are similar in terms of community composition.

We study the metabolic potential of the microbiome of each NRCD group identifying 474 metaCyc metabolic pathways. After STAMP analysis (see Materials and Methods), we found that the gut microbiome from patients of the low-NRCD group exhibits statistically significant differences in metabolic pathways related to amino acid biosynthesis (L-tryptophan biosynthesis, super pathway of L-lysine, L-threonine, and L-methionine biosynthesis II) and nitrogen metabolism pathways (urea cycle and L-citrulline biosynthesis), whereas patients from the high-NRCD group exhibit an increase in the myo-inositol degradation pathway (Fig. 1E).

## Microbial co-occurrence networks from tCD patients are more interconnected and diverse

To comprehensively explore the microbial community structure, sparse inverse covariance estimation and model selection techniques were employed to construct co-occurrence networks. We evaluated the topological properties of microbial networks from each NRCD group and the tCD revealing distinct ecological dynamics (Table 4).

The network from tCD patients was the largest, with 435 edges and 231 nodes, reflecting a highly interconnected and diverse microbial community. In contrast, the networks from low-NRCD patients (259 edges and 191 nodes) and high-NRCD patients (277 edges and 186 nodes) were smaller and less complex, indicating that subsets of taxa might have specific ecological roles. Connectivity and cohesiveness vary across networks. The tCD patient network had the shortest average path length (2.955), promoting faster interactions, while networks from low-NRCD patients (4.848) and high-NRCD patients (4.016) were more fragmented. The percentage of positive edges, reflecting cooperative interactions, was highest in the tCD patient network (95.40%) but decreases in low-NRCD (94.21%) and high-NRCD (90.61%), suggesting more competitive interactions in those networks. Modularity, which captures the community structure, was highest in low-NRCD (0.732) and high-NRCD (0.706) groups, indicating well-defined subgroups, while the tCD patient network (0.574) was more integrated.

We also analyzed node-level properties, focusing on those critical for defining keystone taxa—organisms that play pivotal roles in maintaining community structure and functionality (30). Among these properties, betweenness centrality and degree were prioritized due to their relevance in identifying taxa with a significant influence over community dynamics (30). Our analysis revealed distinct keystone taxa across the three networks, underscoring differences in ecological roles and network architecture (Fig. 2A, Table 4).

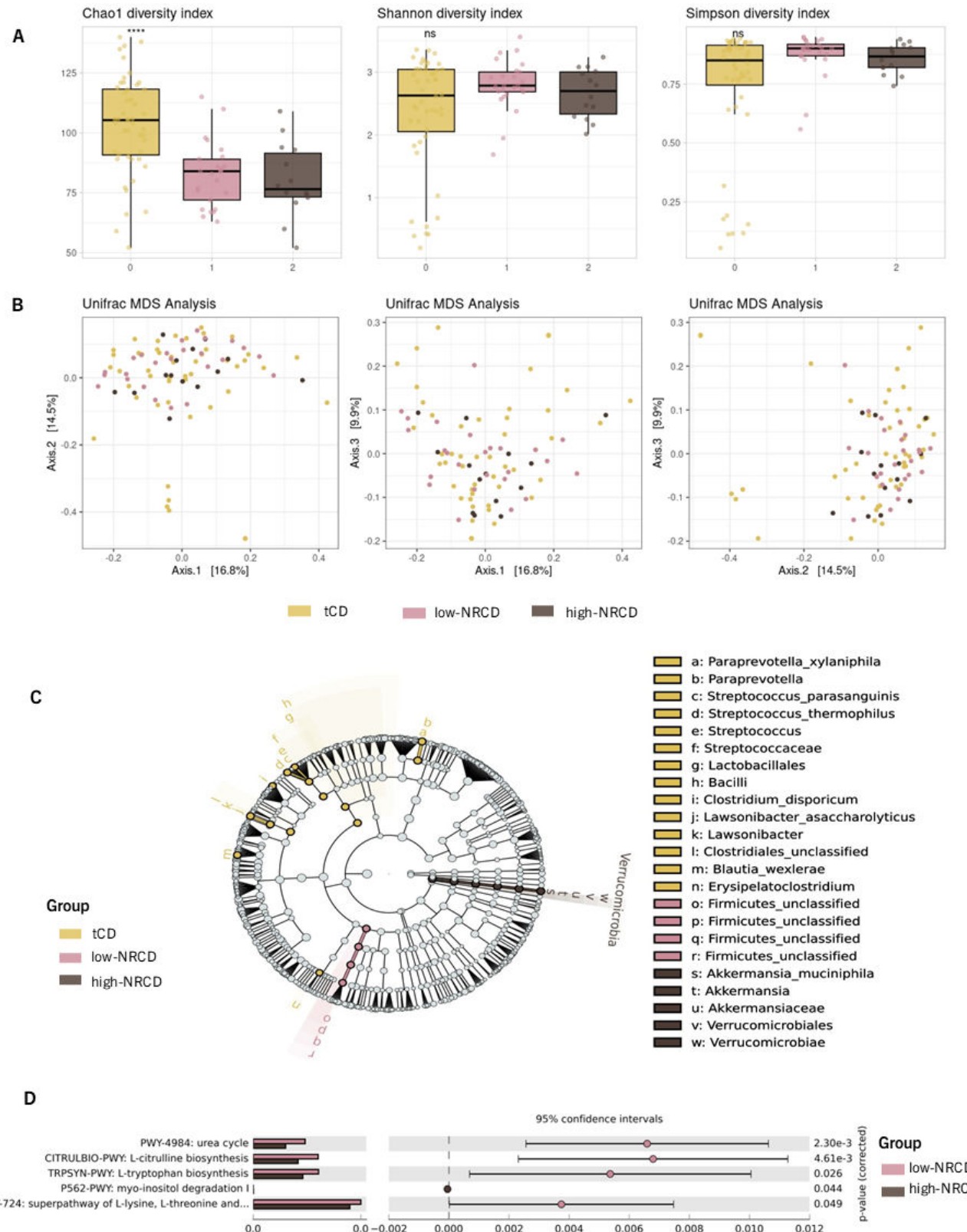

FIG 1 Microbial community structure analysis: (A) Alpha diversity measured by Chao1, Shannon, and Simpson diversity indexes. (B) Multidimensional scaling (MDS) of the Unifrac distance representing beta diversity of the samples. (C) Cladogram showing the taxonomic relations of the microbial markers identified after linear discriminant analysis effect size (LEfSe) analysis using an LDA score >3. (D) MetaCyc pathways differing significantly between NRCD patients with low-grade symptomatology (low-NRCD) and high-grade symptomatology (high-NRCD) with an effect size ≥0.75%.

**TABLE 4** Network topological properties of the microbial co-occurrence networks from asymptomatic treated CD patients (tCD), NRCD patients with low-grade symptomatology (low-NRCD), and NRCD with high-grade symptomatology (high-NRCD)

| Network properties | tCD patients | Low-NRCD | High-NRCD |
|---|---|---|---|
| Size | 435 | 259 | 277 |
| Order | 231 | 191 | 186 |
| Edge density | 0.017 | 0.014 | 0.016 |
| Clustering coefficient | 0.121 | 0.137 | 0.112 |
| Average shortest path length | 2.955 | 4.484 | 4.016 |
| Positive edge % | 95.402 | 94.208 | 90.613 |
| Modularity | 0.574 | 0.732 | 0.706 |
| Keystone taxa[a] | *Bacteroides finegoldii*, *B. fluxus*, *Bacteroides* sp. CAG633, *Barnesiella intestinihominis*, *Eubacterium callanderi*, *Firmicutes bacterium* CAG145, *Klebsiella variicola*, *Megamonas funiformis*, *Parasutterella eubacterexcrementihominis*, *Pseudoflavonifractor* sp. An184, and *Veillonella parvula* | *Anaerotruncus* sp. CAG528 and *Citrobacter youngae* | *Alistipes finegoldii*, *Eggerthella lenta*, *Escherichia coli*, *Eubacterium rectale*, and *Fusicatenibacter saccharivorans* |

[a]Keystone taxa calculated as nodes with betweenness centrality and degree greater than log-normal quantile 0.90.

The network from tCD patients exhibited 11 keystone taxa, including *Bacteroides finegoldii*, *Barnesiella intestinihominis,* and *Eubacterium callanderi,* among others. The abundance of keystone taxa in the tCD network highlights a more interconnected and potentially resilient microbial community, reflecting a balanced and functionally diverse ecosystem. In contrast, the low-NRCD patient network was characterized by only two keystone taxa: *Anaerotruncus* sp. CAG 528 and *Citrobacter youngae.* These taxa may play specialized roles, but they suggest a community with fewer redundancies and a higher dependency on specific organisms for maintaining network integrity. Similarly, the high-NRCD network included five keystone taxa: *Eggerthella lenta, Fusicatenibacter saccharivorans, Alistipes finegoldii, Escherichia coli*, and *Eubacterium rectale.* These taxa are associated with short-chain fatty acid production, amino acid metabolism, and other metabolic pathways essential for host health. However, their reduced number compared to the tCD patient network suggests a less diverse and potentially more vulnerable community.

Finally, we constructed a differential correlation network in order to evaluate the taxa whose associations change significantly between the two clusters by using Fisher's z-test. We found nine microbial species differentially associated between NRCD groups. In the low-NRCD group, a robust positive association was observed among *Bacteroides xylanisolvens, B. salyersiae, Coprobacter fastidosus,* and *Roseburia hominis,* indicating that these species tend to co-occur in the majority of microbiomes from patients exhibiting low-grade symptoms. Conversely, in the high-NRCD group, a positive association was identified between *B. xylanisolvens, B. thetaiotaomicron, Ruminococcus torques*, and *Dorea longicatena* on one hand and *B. eggerti* and *Prevotella* sp. CAG1031 on the other hand. These microbial association disparities suggest variations in the community structure and interactions present therein (Fig. 2B).

## Microbial co-occurrence networks from patients with NRCD are less stable than microbial networks from asymptomatic patients

We assessed network robustness by evaluating how resilient each network is to the loss of hubs (nodes with high degree) and bottlenecks (nodes with high betweenness centrality). To measure robustness, we sequentially removed nodes based on their degree and betweenness centrality and recorded the size of the resulting graph's largest connected component (LCC).

When nodes were removed based on the degree (Fig. 2F), the tCD patients network could withstand the removal of more nodes before becoming disconnected, compared to the low-NRCD and high-NRCD patients' networks. This indicates that the tCD patients' network is structurally more robust to targeted attacks on highly connected nodes. A similar trend was observed when nodes were removed based on betweenness centrality

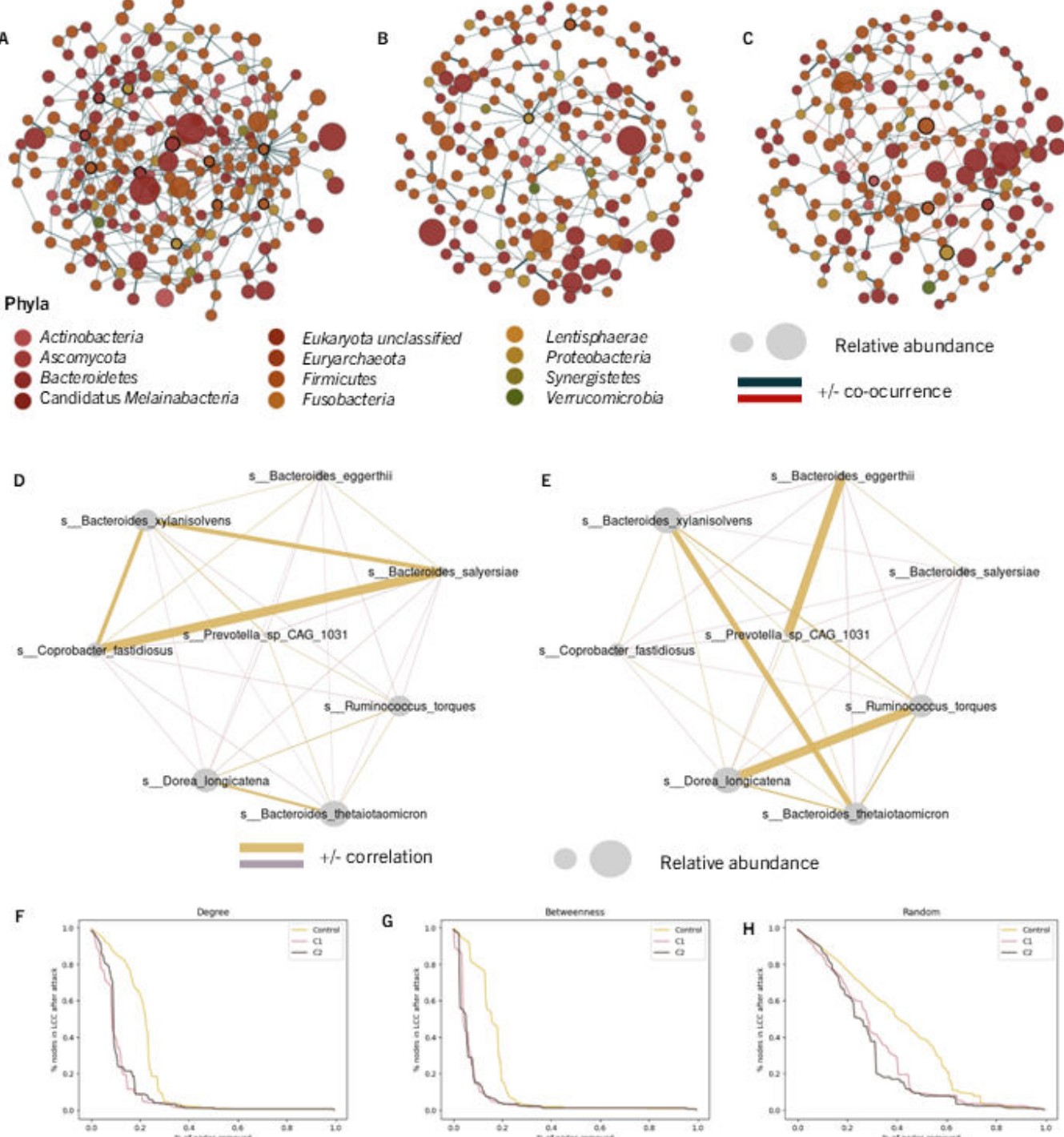

**FIG 2** Microbial network analysis: Co-occurrence networks of the microbiome from (A) asymptomatic treated CD patients (tCD). (B) NRCD patients with low-grade symptoms (low-NRCD). (C) NRCD patients with high-grade symptoms (high-NRCD). (D) Network of differentially associated species in NRCD patients with low-grade symptoms (low-NRCD). (E) Network of differentially associated species in NRCD patients with high-grade symptoms (high-NRCD). (F) Number of nodes in the networks' LCC compared to the original after sequential removal of nodes with the highest degree. (G) Same as panel F, but with removal of nodes based on their highest betweenness centrality. (H) Same as panel F, but with removal of random nodes. The nodes in the graph represent different bacterial species. The size of each node corresponds to its relative abundance, while the edges represent statistically significant associations between the nodes ($P < 0.05$). Green edges indicate positive relationships, while red edges indicate negative ones. The thickness of each edge indicates the strength of the association. Nodes highlighted in panels A, B, and C correspond to keystone taxa calculated as nodes with betweenness centrality and degree greater than log-normal quantile 0.90.

(Fig. 2G), further emphasizing the greater resilience of the tCD patients' network. We also evaluated the networks' resistance to random attacks (Fig. 2H), where nodes were removed at random, without considering their topological properties. The tCD patient network demonstrated high resilience to these random removals, with the size of the LCC (i.e., the number of remaining connected nodes) decreasing almost linearly as nodes were removed. In contrast, low-NRCD and high-NRCD networks exhibited a more rapid decline in connectivity under similar conditions, reflecting their reduced structural stability compared to the tCD network. These findings underline the greater robustness of the tCD patients' network, likely due to its higher connectivity and keystone taxa diversity, as previously shown, which provide a more stable and adaptable microbial ecosystem.

## Microbial-derived metabolites differed between the two groups of NRCD patients

A total of 372 metabolites were identified by HPLC (Data S3) and classified into four groups: five host (human)-specific metabolites, 56 microbial metabolites, 103 microbial–host cometabolites, and 208 others (57 drug, 48 food, 1 environmental related, and 102 unknown). There were 36 differential metabolites between the two clusters, including 6 microbial metabolites, 12 host–microbial co-metabolites, and 18 others. A metabolic pathway enrichment analysis (MPEA) was performed with the microbial and host–microbial cometabolites (Fig. 3). There were 6 and 29 related metabolic pathways that could be matched against microbial and host–microbial metabolites pathway databases, respectively. Among these, pathways showing significant differences between clusters ($P < 0.05$) were lysine biosynthesis (microbial) and glutathione metabolism, glyoxylate and dicarboxylate metabolism, C5-branched dibasic acid metabolism, alanine, aspartate, and glutamate metabolism, pentose phosphate pathway, butanoate metabolism, D-amino acid metabolism, and arginine and proline metabolism (co-metabolism).

Through Sankey network analysis, we were able to link all bacteria that might participate in a specific metabolic reaction, allowing us to identify the interplay between bacteria and metabolites and the biological correlation between bacteria and metabolites (Data S4).

We constructed an integrated network comprising the metabolic pathways identified post-MPEA, which exhibited significant differences between patient clusters, alongside associated metabolites and microorganisms that had been validated through biological and statistical correlations via Sankey network analysis (Fig. 4). Our findings reveal that two microbial species, *A. hadrus* and *L. assacharolitycus,* enriched in patients from the high-NRCD and low-NRCD groups, respectively, were associated with nine metabolites implicated in these metabolic pathways.

Specifically, within the lysine biosynthesis metabolic pathway, we observe that 2,6-diaminopimelic acid showed a positive correlation, and its abundance was positively associated with both microbial species. Furthermore, we observed that the metabolic pathways mainly related to amino acid metabolism (ko00470, ko00330, ko00250, ko00480, and ko00300) were positively correlated with a series of metabolites (aconitic acid, glyceric acid, pyroglutamic acid, 1,5-pentadiamine, glutamic acid, gamma-aminobutyric acid, 2-deoxyribose, and glutamine) overrepresented in patients from the high-NRCD group and positively correlated with *A. hadrus*. Meanwhile, the presence of *L. assacharolitycus* was negatively correlated with the presence of glutamic acid and 1,5-pentadiamine, which in turn play a significant role in regulating the metabolic pathways of glutathione, arginine, and proline metabolism (ko00480, ko00470, and ko00330).

In summary, the host-specific metabolic and microbe-metabolite association networks provide more specific information on metabolic changes observed in patients from the two groups, highlighting the metabolites related to these metabolic pathways and their regulation by microbial species.

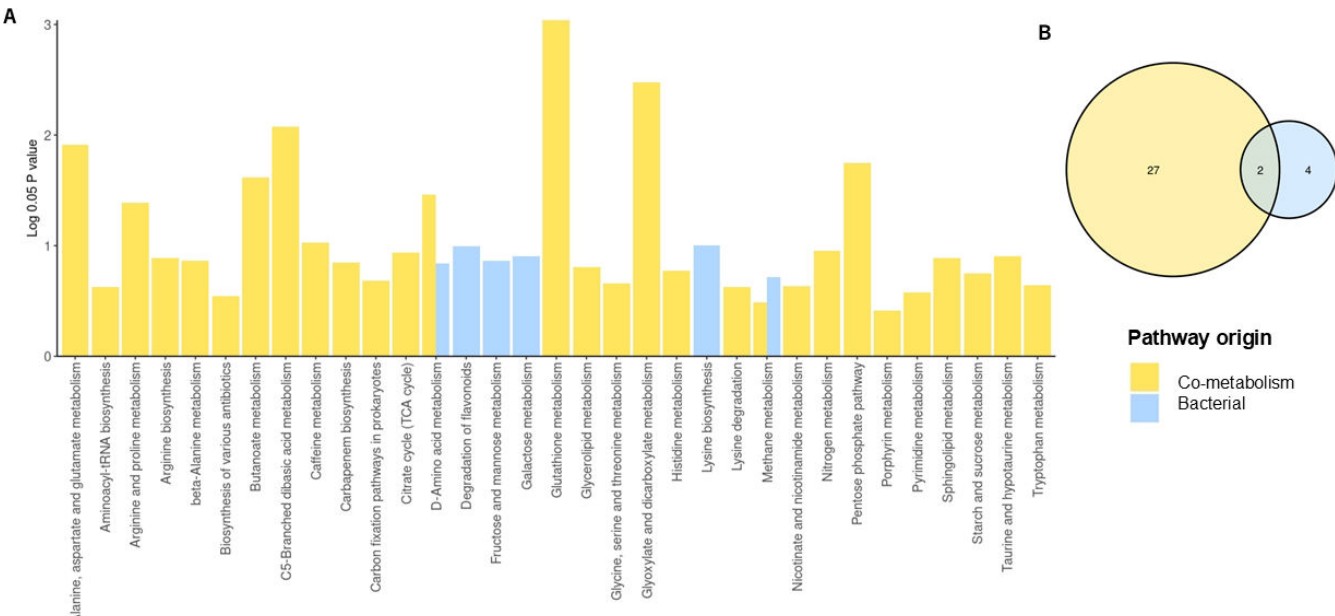

FIG 3   (A) Metabolite pathway enrichment analysis (MPEA) of microbiota-derived (bacterial) and co-metabolism-derived metabolites. (B) Number of pathways of bacterial and host co-metabolism origin. Bar plot showing the relative significance of differential metabolic pathways from MPEA according to the differential metabolites from bacterial and host co-metabolism origins.

## Association between symptom severity and fecal microbiome and metabolome

The potential association between changes in symptom scores and changes in gut microbiome (keystone and differentially associated taxa), microbial functional potential (differentially abundant metaCyc pathways), and fecal metabolome (differentially abundant metabolites) were assessed by performing Pearson's correlation test (Fig. 5).

Regarding gut microbiome composition and functional potential, a positive association was observed between the presence of nausea and the microorganisms: *Ruminococcus torques*, *Citrobacter youngae,* and *Dorea longicatena*. The latter also exhibited a positive correlation with the total score of the CeD-Pro questionnaire. Moreover, the microbial metabolic pathway P562-PWY (myo-inositol degradation I) showed a positive association with the presence of flatulence and diarrhea, along with the total score of the CeD-Pro questionnaire.

Regarding the fecal metabolome, we identified the following associations. (i) For metabolites associated with energy metabolism: a significant positive association was found between glyceric acid levels and the presence of loose stool and diarrhea, mannitol levels positively correlated with nausea and loose stools, 2-hydroxyisobutyric acid levels were linked to nausea, and finally, 2-deoxyribose was positively correlated to loose stool, diarrhea, and the overall GSRS questionnaire score. (ii) For metabolites linked to amino acid and protein metabolism: glutamic acid was correlated with diarrhea and total and CD-specific GSRS questionnaire scores. N-methylalanine was correlated with loose stools, gastrointestinal pain or discomfort, and total scores on GSRS and Ce-DPro questionnaires. Finally, 2-propanamidoacetic acid was associated with loose stool. (iii) For metabolites related to pH regulation and intestinal function: pyroglutamic acid was positively associated with loose stools, gastrointestinal pain or discomfort, and total and partial GSRS questionnaire scores. Similarly, indolelactic acid was linked to urgent bowel movement, loose stools, diarrhea symptoms, and total and partial scores on GSRS and CeD-Pro questionnaires. Additionally, galactonic acid positively correlated with loose stool and total and partial scores on GSRS and CeD-Pro questionnaires. The aforementioned metabolites are found to be more abundant in patients from the

high-NRCD group, underscoring the potential role of these metabolites in contributing to the severity and/or persistence of symptoms.

## DISCUSSION

The gut microbiome has garnered significant attention in the context of active CD, where its pivotal role, particularly a dysbiosis marked by an abundance of gram-negative bacteria like *Bacteroidetes* and *Proteobacteria*, has been extensively explored (11). However, in the case of NRCD, the scientific understanding remains relatively limited. Nevertheless, emerging evidence hints at the potential influence of small intestinal bacterial overgrowth (SIBO) on the persistence of symptoms in these patients (6, 8, 9). While this area requires further investigation, it underscores the relevance of exploring the gut microbiome's involvement in the persistence of symptoms in CD patients despite maintaining a GFD.

Our study provides novel insights into NRCD, identifying two distinct symptom-based patient clusters: "Low-grade symptoms: Cluster 1 (low-NRCD)" and "High-grade symptoms: Cluster 2 (high-NRCD)." These groups exhibited differences in symptoms, inflammatory markers, microbial composition, and associations between microbial metabolites and symptoms. Notably, both groups demonstrated increased intestinal permeability, as evidenced by elevated lactulose:mannitol ratios, despite preserved enterocyte functionality (normal I-FABP and fecal calprotectin levels). These findings suggest that factors beyond enterocyte function, such as microbial dynamics, contribute to ongoing symptoms and their interplay with membrane permeability and inflammation.

### Microbial composition and co-occurrence networks

Analyzing the microbial composition in these groups and comparing it to asymptomatic treated CD (tCD) patients revealed limited differences in global microbial diversity. However, microbial co-occurrence network analysis offered a deeper understanding of community structure and dynamics. This approach highlighted keystone taxa—

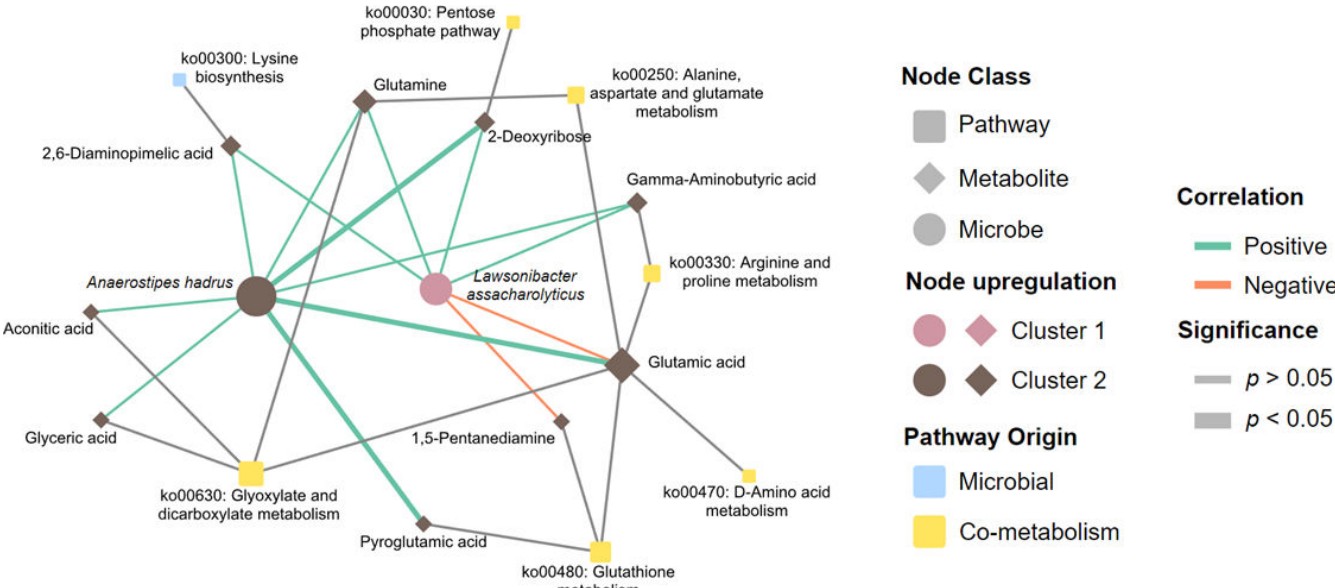

**FIG 4** Network integrating microbiome and metabolome interactions. Differential metabolites from microbiota and co-metabolism origins and their related bacteria and involved KEGG metabolic pathways are shown. Nodes represent bacterial species, metabolites, or pathways. The three kinds of nodes are differentiated by node shape. Node size is proportional to node degree. Microbe and metabolite nodes are colored depending on the cluster in which they are upregulated. Pathway node color represents whether it is present only in microbial or also in host metabolism. Edges between microbes and metabolites are colored based on the correlation sign. Edges representing significant correlations ($P < 0.05$) are represented by a thicker line.

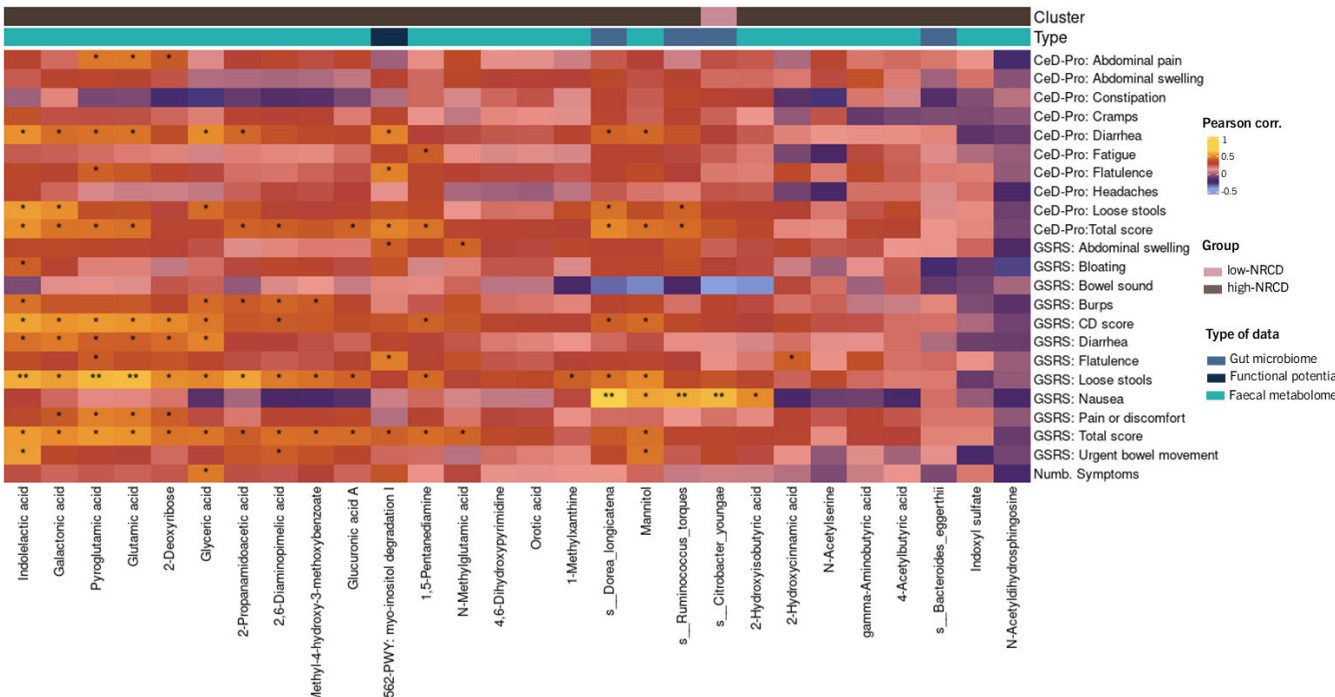

**FIG 5** Heatmap representation of Pearson's correlation between changes in symptom scores and changes in gut microbiome (keystone and differential associated taxa), microbial functional potential (differential abundance metaCyc pathways), and fecal metabolome (differentially abundant metabolites). Cluster annotation represents if the value is more abundant in the low-NRCD group (pink) or high-NRCD group (brown). CeD-Pro: Celiac Disease Patient-Reported Outcome. GSRS: Gastrointestinal Symptom Rating Scale Celiac Disease. Significant results: (*) $P < 0.05$; (**) $P < 0.01$.

organisms central to ecosystem stability—often missed by conventional diversity metrics (30).

In tCD patients, the microbial network exhibited a compact structure with numerous keystone taxa, including *Bacteroides finegoldii, Barnesiella intestinihominis,* and *Eubacterium callanderii*, among others. These taxa are associated with critical functions such as fiber and complex polysaccharide metabolism, gut barrier maintenance, and immunomodulation through production of short-chain fatty acids (SCFAs) (38–41). The network's resilience, evident from its stability against random and directed node removal (connectivity maintained until ~40% of nodes were eliminated), and the high proportion of positive interactions (95.33%) reflect a cooperative and robust ecosystem.

In low-NRCD patients, the network displayed a fragmented structure with fewer keystone taxa, notably *Anaerotruncus sp. CAG528* and *Citrobacter youngae*. Despite their limited number, these taxa perform specialized roles, such as SCFA production, which is critical for maintenance of gut health (42). The high modularity and longer average path length of this network suggest localized clustering under stress, relying on a few critical taxa to maintain connectivity. However, a distinction emerges upon comparative analysis with the high-NRCD network, revealing an intriguing association involving a subgroup of *Bacteroides*. Previously identified because of their probiotic potential, these particular taxa (*B. xylanisolvens* and *B. salyersiae*) are recognized as producers of SCFAs, primarily acetate and propionate (43–45). The co-occurrence of these *Bacteroides* species is observed to enhance the presence of *Coprobacter fastidiousus* and *Roseburia hominis*. Notably, the latter has been linked to the strengthening of gut barrier function and possesses immunomodulatory properties, suggesting its potential role in the control and treatment of gut inflammation (46). These findings suggest the presence of a microbial community associated with the homeostasis of the immune response and the maintenance of barrier integrity, lowering the severity of inflammation and gastrointestinal symptoms.

The high-NRCD group, associated with severe symptoms, had a more interconnected yet antagonistic network. Keystone taxa included *Eggerthella lenta*, *Fusicatenibacter saccharivorans*, *Alistipes finegoldii*, *Escherichia coli*, and *Eubacterium rectale*. These taxa play dual roles: SCFA producers like *E. rectale* and *F. saccharivorans* regulate immune responses (47, 48), while lipopolysaccharide (LPS) producers (*E. coli* and *A. finegoldii*) and Th17 activators (*E. lenta*) exacerbate inflammation (49–51). This network's higher edge density and lower proportion of positive interactions (90.61%) suggest increased competition and immune dysregulation. Moreover, differential association network analysis revealed significant shifts in microbial interactions. In the high-NRCD network, *B. xylanisolvens* and *B. thetaiotaomicron* were positively associated with *Ruminococcus torques* and *Dorea longicatena*, microorganisms previously linked to compromised gut barrier integrity and symptom severity in IBD (52). This contrasts with the robust associations observed in low-NRCD patients, suggesting a shift from a cooperative to a more antagonistic microbial community structure in high-NRCD patients.

These findings underscore the functional implications of microbial interactions, highlighting the potential of specific taxa to influence host health and symptom severity. While the low-NRCD network reflects a community capable of mitigating inflammation and supporting gut homeostasis, the antagonistic interactions in the high-NRCD network may amplify the inflammatory state, providing potential targets for therapeutic interventions.

## Immune and metabolic implications

Patients in the high-NRCD group exhibited elevated pro-inflammatory cytokine (IL-12 and TNFα) levels alongside pronounced vitamin D deficiency. This deficiency, more marked in severe cases, highlights the nutrient's potential role in mitigating inflammation and preserving gut integrity. Vitamin D's immunomodulatory properties and ability to protect the intestinal mucosa from damage underscore its relevance in CD pathogenesis and symptom severity (53). Evidence indicates that vitamin D plays a protective role in preserving the intestinal mucosa from chemical and immunological damage. Moreover, its modulation of immune system functions can counteract the mechanisms underlying intestinal modifications typical of gastrointestinal autoimmune diseases (53). Hence, the observed higher deficiency of vitamin D in patients experiencing more pronounced symptoms suggests a potential association between the extent of symptoms and the role that this nutrient may play in symptom manifestation.

Previous research has established that CD is characterized by metabolic alterations encompassing impaired energy metabolism, malabsorption, and perturbations in the gut microbiota (54). In our metabolomic analysis, we aimed to distinguish between microbial- and host-derived metabolites, as well as co-metabolites resulting from their interactions. Notably, patients belonging to the high-NRCD group exhibited elevated levels of metabolites associated with tryptophan metabolism (indoxyl sulfate and indolelactic acid), protein hydrolysis (cadaverine), bacterial cell wall components (2,6-diaminopimelic acid), and inflammation (mannitol). Tryptophan metabolism plays a pivotal role in the complex interplay of the microbiota–gut–brain axis. Serving as the exclusive precursor for the neurotransmitter serotonin, it holds essential significance in the regulation of emotional processing, pain perception, as well as the modulation of colonic motility and secretory activity within the gastrointestinal system (55). Our results showed that indolelactic acid, higher in patients from the high-NRCD group, was positively correlated with gastrointestinal symptoms and partial and total scores in the CD-related questionnaires, showing an association between this microbial-derived metabolite and symptom persistence.

We gained insights into the intricate interplay between the metabolome and microbiome by integrating the metabolic pathways of the microbiome with the metabolites detected in the samples. Our findings revealed that pathways mainly related to amino acid metabolism were enriched, and two microbial species (*A. hadrus* and *L. asaccharolyticus*) were significantly related to the metabolites participating in those

pathways. Abnormalities in amino acid metabolism have been associated with impairment in gut barrier integrity through activation of the immune response by Th17 activation (51).

## Linking microbial metabolism to symptoms

Furthermore, we conducted a comprehensive investigation into the association between microbial taxa, differential metabolites, metabolic pathways, and the observed symptoms. Our analysis revealed a strong association between microbial-derived metabolites (mannitol, indolelactic acid, and galactonic acid), human–microbial co-metabolism-derived metabolites (2-deoxyribose, glutamic acid, pyroglutamic acid, and glyceric acid) and gastrointestinal symptoms, showing that higher levels of these metabolites are associated with higher scores in partial and total CD-related questionnaires.

We also found that the myo-inositol degradation pathway was significantly more abundant in patients from the high-NRCD group ($P < 0.005$) and was associated with the presence of flatulence, diarrhea, and the summary of symptoms from the questionnaires. Myo-inositol degradation is essential for carbon assimilation in bacteria, and it serves as a key player in various metabolic functions. Inadequate myo-inositol levels have been linked to heightened inflammation and apoptosis in the intestinal mucosa, along with reduced cell proliferation and antioxidant capacity. This pathway exhibits activity exclusively in specific kidney cell types and becomes especially relevant under certain disease conditions (56, 57). Here we show, for the first time, its possible association with CD symptoms based on its relevance in the high-NRCD group.

Prior research has established a correlation between alterations in the gut microbiome composition and the occurrence or intensification of various gastrointestinal symptoms (58). Our findings support the hypothesis that the metabolic capabilities of the microbiome, specifically reflecting the functional aspects of the microbial community, exert a significant influence on the persistence of symptoms. We hypothesize that this influence may be primarily mediated through the regulation of amino acid metabolism and immune homeostasis led by Th17 response. Of particular importance, we underscore the involvement of specific microorganisms and their associated metabolites in triggering immune activation, which subsequently leads to alterations in the epithelial barrier. These changes in the barrier function may give rise to visceral hypersensitivity and abnormalities in gut motility, ultimately manifesting as severe symptomatology.

## Conclusion

Our findings support the hypothesis that persistent symptoms in NRCD are driven by disruptions in microbiome composition and function. These disruptions affect amino acid metabolism, immune balance (via Th17 responses), and epithelial barrier integrity. Differences in microbial community organization—ranging from robust ecosystems in tCD patients to fragmented, stress-prone networks in symptomatic clusters—highlight the role of microbial interactions in shaping disease outcomes. By identifying specific taxa, metabolites, and pathways associated with symptom persistence, this study provides a framework for future investigations into therapeutic targets for NRCD patients.

## ACKNOWLEDGMENTS

This study has been funded by CD3DTech-CM (TEC-2024/BIO-167), a Research Grant 2020 of the European Society of Clinical Microbiology, Infectious Diseases (ESCMID) to L.J.M.-Z; Project PID2023-150146OA-I00 founded by MICIU/AEI /10.13039/501100011033 and FEDER, UE; Institute of Health Carlos III (project IMPaCT-Data, exp. IMP/00019), co-funded by the European Union, European Regional Development Fund ('A way to make Europe'); AI4FOOD-CM (Y2020/TCS-6654), RED2022-134934-T funded by MICIU/AEI/

10.13039/501100011033. B.L.P is funded by Formación del ProfesoradoUniversitario grant (FPU22/04053) from the Spanish State Ministerio de Universidades.

## AUTHOR AFFILIATIONS

[1]Computational Biology Group, IMDEA Food, CEI UAM+CSIC, Madrid, Spain

[2]UAM Doctoral School, Universidad Autónoma de Madrid, Madrid, Spain

[3]GENYAL Platform, IMDEA Food, CEI UAM+CSIC, Madrid, Spain

[4]Department of Pharmacy and Nutrition, Faculty of Biomedical and Health Sciences, Universidad Europea de Madrid, Villaviciosa de Odón, Spain

[5]Foodomics Lab, Institute of Food Science Research (CIAL, CSIC), Madrid, Spain

[6]Department of Nutrition and Food Science, Faculty of Pharmacy, VALORNUT Research Group, Complutense University of Madrid, Madrid, Spain

[7]Laboratory of Cellular and Molecular Gerontology, IMDEA Food, CEI UAM+CSIC, Madrid, Spain

[8]CIBER Physiopathology of Obesity and Nutrition (CIBERobn), Madrid, Spain

[9]Centre de Recherches en Cancérologie de Toulouse, CRCT, Université de Toulouse, Inserm, CNRS, Toulouse, Occitanie, France

## PRESENT ADDRESS

Elena Aguilar-Aguilar, Department of Pharmacy and Nutrition, Faculty of Biomedical and Health Sciences, Universidad Europea de Madrid, Villaviciosa de Odón, Madrid, Spain

Helena Marcos-Pasero, Department of Pharmacy and Nutrition, Faculty of Biomedical and Health Sciences, Universidad Europea de Madrid, Villaviciosa de Odón, Madrid, Spain

Viviana Loria-Kohen, Department of Nutrition and Food Science, Faculty of Pharmacy, VALORNUT Research Group, Complutense University of Madrid, Madrid, Spain

## AUTHOR ORCIDs

Laura Judith Marcos-Zambrano  http://orcid.org/0000-0003-1381-6407

## FUNDING

| Funder | Grant(s) | Author(s) |
|---|---|---|
| Comunidad de Madrid | TEC-2024/BIO-167 CD3DTech-CM (ORDER 5696/2024, B.O.C.M. No. 307 12/26/2024) | Laura Judith Marcos-Zambrano |
|  |  | Elena Aguilar-Aguilar |
|  |  | Helena Marcos-Pasero |
|  |  | Alberto Diaz-Ruiz |
|  |  | Enrique Carrillo de Santa Pau |

## AUTHOR CONTRIBUTIONS

Laura Judith Marcos-Zambrano, Conceptualization, Formal analysis, Funding acquisition, Writing – original draft | Blanca Lacruz-Pleguezuelos, Formal analysis, Visualization | Elena Aguilar-Aguilar, Data curation, Formal analysis, Investigation | Helena Marcos-Pasero, Data curation, Formal analysis, Investigation | Alberto Valdés, Formal analysis, Investigation | Viviana Loria-Kohen, Funding acquisition, Investigation, Methodology, Resources, Writing – review and editing | Alejandro Cifuentes, Formal analysis, Resources | Ana Ramirez de Molina, Resources, Writing – review and editing | Alberto Diaz-Ruiz, Resources, Writing – review and editing | Vera Pancaldi, Funding acquisition, Investigation, Resources, supervision, Writing – review and editing | Enrique Carrillo de Santa Pau, Funding acquisition, Investigation, Resources, supervision, Writing – review and editing

## DATA AVAILABILITY

The data sets generated and analyzed during the current study are available in the ENA repository (PRJEB65879). The metabolomics data are available within the paper. A full

metadata record, data analysis, and original R scripts are available on GitHub at https://github.com/laurichi13/NRCD_analysis.

## ETHICS APPROVAL

The study was approved by the Research Ethics Committee of the IMDEA Food Foundation (PI-032; approval date: 12 June 2017). All the participants have signed the informed consent and donated the samples for future research in the same line of research. These samples are a collection: Reference: C.0004841; Registro Nacional de Biobancos Hospital Carlos III; PI: Dr. Viviana Loria Kohen; date: 21 March 2019.

## ADDITIONAL FILES

The following material is available online.

### Supplemental Material

**Data S1 (mSystems00143-25-s0001.xlsx).** STORMS checklist.
**Data S2 (mSystems00143-25-s0002.xlsx).** HPLC results.
**Data S3 (mSystems00143-25-s0003.xlsx).** Metagenomics' comparisons.
**Data S4 (mSystems00143-25-s0004.xlsx).** Metabolomics' comparisons.
**Supplemental Figures (mSystems00143-25-s0005.docx).** Figures S1 to S3
**Figure S4 (mSystems00143-25-s0006.pdf).** Sankey network diagram.
**Supplemental legends (mSystems00143-25-s0007.docx).** Legends for supplemental material.

### Open Peer Review

**PEER REVIEW HISTORY (review-history.pdf).** An accounting of the reviewer comments and feedback.

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
