## [Reviewer comments · mSystems]

Microbiome gut community structure and functionality is associated with symptoms severity in non-responsive celiac disease patients undergoing gluten-free diet.

Laura Marcos-Zambrano, Blanca Lacruz-Pleguezuelos, Elena Aguilar-Aguilar, Helena Marcos-Pasero, Alberto Valdés, Viviana Loria-Kohen, Alejandro Cifuentes, Ana Ramirez de Molina, Alberto Díaz-Ruiz, Vera Pancaldi, and Enrique Carrillo de Santa Pau

Corresponding Author(s): Laura Marcos-Zambrano, Instituto Madrileño de Estudios Avanzados en Alimentación

Review Timeline:

Submission Date:	January 31, 2025
Editorial Decision:	February 27, 2025
Revision Received:	March 12, 2025
Accepted:	March 25, 2025

Editor: Marc Cook

Reviewer(s): Disclosure of reviewer identity is with reference to reviewer comments included in decision letter(s). The following individuals involved in review of your submission have agreed to reveal their identity: Yan Ni (Reviewer #1); Jing Wu (Reviewer #2)

Transaction Report:

DOI: <https://doi.org/10.1128/msystems.00143-25>

Re: mSystems00143-25 (Microbiome gut community structure and functionality is associated with symptoms severity in non-responsive celiac disease patients undergoing gluten-free diet.)

Dear Dr. Laura Judith Marcos-Zambrano:

Revision Guidelines

Sincerely,
Marc Cook
Editor
mSystems

Reviewer #2 (Comments for the Author):

We appreciate the authors' efforts to promote their work.

There are three concerns that require explanation and two recommendations:

1.The public data of asymptomatic tCD from "Francavilla A, Ferrero G, Pardini B, Tarallo S, Zanatto L, Caviglia GP, et al. Gluten-free diet affects fecal small non-coding RNA profiles and microbiome composition in celiac disease supporting a host-gut microbiota crosstalk. Gut Microbes. 2023;15:2172955."

The original tCD patients were 51(Fifty-one tCD individuals were on a strict GFD and tested negative for TG IgA antibodies (here after defined as tCD-TG-) at recruitment.). But the authors of this article only included 48. Please explain the reasons and clarify the selection criteria in the final version of article.

Please provide supplementary documents to demonstrate the comparison. Please inform the readers if there is any comparison on the metabolites.

2.Different diet(Italian and Spanish) will affect the metabolites. Please remind the readers if the authors could not provide the data after correction by using algorithm.

3.Figure 5 is quite different from previous version figure 4, especially the first line of cluster information.Please provide reasonable explanation. Please differentiate the gut microbiome (keystone and differential associated taxa), microbial functional potential (differential abundance and metaCyc pathways), and faecal metabolome (differentially abundant metabolites) .

4.In article line 576 "There were 36 differential metabolites between the two clusters, including 6 microbial metabolites, 12 host-microbial co-metabolites, and 18 others."

Please provide the complete supplementary documents to demonstrate the comparison,including the FDR.

5.The authors claimed the relation between Th 17 and their results. It will be too early to make "These disruptions affect amino acid metabolism, immune balance (via Th17 responses), and epithelial barrier integrity. ", since there are no evidence(e.g. IL-17) for them to support their conclusion. It is just a possibility.

We appreciate the authors' efforts to promote their work.

There are three concerns that require explanation and two recommendations:

1.The public data of asymptomatic tCD from "Francavilla A, Ferrero G, Pardini B, Tarallo S, Zanatto L, Caviglia GP, et al. Gluten-free diet affects fecal small non-coding RNA profiles and microbiome composition in celiac disease supporting a host-gut microbiota crosstalk. Gut Microbes. 2023;15:2172955."

The original tCD patients were 51(Fifty-one tCD individuals were on a strict GFD and tested negative for TG IgA antibodies (here after defined as tCD-TG-) at recruitment.). But the authors of this article only included 48. Please explain the reasons and clarify the selection criteria in the final version of article.

Please provide supplementary documents to demonstrate the comparison. Please inform the readers if there is any comparison on the metabolites.

2.Different diet(Italian and Spanish) will affect the metabolites. Please remind the readers if the authors could not provide the data after correction by using algorithm.

3.Figure 5 is quite different from previous version figure 4, especially the first line of cluster information.Please provide reasonable explanation. Please differentiate the gut microbiome (keystone and differential associated taxa), microbial functional potential (differential abundance and metaCyc pathways), and faecal metabolome (differentially abundant metabolites) .

4. In article line 576 “There were 36 differential metabolites between the two clusters, including 6 microbial metabolites, 12 host-microbial co-metabolites, and 18 others.”

Please provide the complete supplementary documents to demonstrate the comparison,including the FDR.

5.The authors claimed the relation between Th 17 and their results. It will be too early to make “These disruptions affect amino acid metabolism, immune balance (via Th17 responses), and epithelial barrier integrity. ” , since there are no evidence(e.g. IL-17) for them to support their conclusion. It is just a possibility.

Reviewer #2 (Comments for the Author):

We appreciate the authors' efforts to promote their work. There are three concerns that require explanation and two recommendations:

1. The public data of asymptomatic tCD from "Francavilla A, Ferrero G, Pardini B, Tarallo S, Zanatto L, Caviglia GP, et al. Gluten-free diet affects fecal small non-coding RNA profiles and microbiome composition in celiac disease supporting a host-gut microbiota crosstalk. Gut Microbes. 2023;15:2172955."

The original tCD patients were 51(Fifty-one tCD individuals were on a strict GFD and tested negative for TG IgA antibodies (here after defined as tCD-TG-) at recruitment.). But the authors of this article only included 48. Please explain the reasons and clarify the selection criteria in the final version of article.

Thank you for your careful review and for pointing out this discrepancy. Upon revisiting the dataset, we confirm that metagenomic data was available for 50 tCD-TG- samples. However, we were unable to retrieve data from samples Cii007 and Cii009 due to file corruption, which rendered them unusable for analysis. As a result, only 48 samples were ultimately included in our study.

We included the following text in the Methods section:

Lines 245 - 248: "Initially, metagenomic data were available for 50 tCD-TG- samples. However, data from two samples were unrecoverable due to file corruption, preventing their inclusion in the analysis. Consequently, the final dataset comprised 48 tCD-TG- samples."

Please provide supplementary documents to demonstrate the comparison.

We include as **Supplementary Data S3** the alpha diversity values computed for each sample, the linear discriminant analysis effect size (LEfse) results showing taxon's table and LDA score from tCD, low-NRCD, and high-NRCD patients, and the differential abundance analysis of each species comparing low-NRCD and high-NRCD patients made with the Mann-Whitney Test.

Please inform the readers if there is any comparison on the metabolites.

Thank you for your inquiry. We do not have metabolite data available for this cohort, so no metabolite comparisons were made. We clarify this in methods section:

Line 294: "Metabolomics data was only available for subjects from IMDEA Food Institute."

2. Different diet(Italian and Spanish) will affect the metabolites. Please remind the readers if the authors could not provide the data after correction by using algorithm.

We acknowledge that we do not have metabolomic data to assess whether dietary differences between Spain and Italy lead to variations in metabolite profiles. Regarding the microbiota, it is well established that dietary changes can influence the gut microbiome. In this study, our primary focus was ensuring that all participants adhered to a gluten-free diet for at least 12

months, which was achieved. However, we recognize that other minor dietary variations may exist, and this is a limitation we acknowledge.

Nevertheless, our statistical approach effectively corrected for potential batch effects between groups, reducing variability that could stem from geographical dietary differences. As shown in Supplementary Figure 3, we successfully minimized this batch effect, allowing for valid comparisons across groups. Therefore, we do not consider it necessary to include a specific discussion on diet type. Additionally, despite cultural differences, both Spain and Italy predominantly follow a Mediterranean diet with some variations, but these are not extreme enough to introduce substantial dietary divergence.

3. Figure 5 is quite different from previous version figure 4, especially the first line of cluster information. Please provide reasonable explanation. Please differentiate the gut microbiome (keystone and differential associated taxa), microbial functional potential (differential abundance and metaCyc pathways), and faecal metabolome (differentially abundant metabolites) .

Thank you for your observation. The figure was modified because we decided to remove the GO term information from the previous version, as these results were derived from the HUMAnN analysis but were neither discussed nor commented on in the manuscript. We felt that they did not add significant value to the study.

To improve clarity, we redesigned the heatmap, explicitly annotating the type of data presented (gut microbiome, microbial functional potential, and metabolites). Additionally, we removed the top dendrogram, as it did not contribute to the interpretation of the figure nor was it discussed in the manuscript; it merely displayed similarities between data groups. We believe that the revised figure is now clearer and more aligned with the results discussed in the manuscript.

4. In article line 576 "There were 36 differential metabolites between the two clusters, including 6 microbial metabolites, 12 host-microbial co-metabolites, and 18 others."

Please provide the complete supplementary documents to demonstrate the comparison, including the FDR.

We include as **Supplementary Data S4** the results of the metabolites' origin analysis, the differential analysis of metabolites from low-NRCD and high-NRCD patients made with the Mann-Whitney Test, including FDR correction, and the metabolic pathway enrichment analysis (MPEA) according to the differential metabolites from co-metabolism. and microbiota.

5. The authors claimed the relation between Th 17 and their results. It will be too early to make "These disruptions affect amino acid metabolism, immune balance (via Th17 responses), and epithelial barrier integrity. ", since there are no evidence (e.g. IL-17) for them to support their conclusion. It is just a possibility.

Thank you for your comment. We acknowledge that the connection between our results and Th17 responses is a hypothesis rather than a direct conclusion. To ensure accuracy, we have revised the statement as follows:

Line 767-769: "We hypothesize that this influence may be primarily mediated through the regulation of amino acid metabolism and immune homeostasis lead by Th17 response."

Re: mSystems00143-25R1 (Microbiome gut community structure and functionality is associated with symptoms severity in non-responsive celiac disease patients undergoing gluten-free diet.)

Dear Dr. Laura Judith Marcos-Zambrano:

Your manuscript has been accepted, and I am forwarding it to the ASM production staff for publication. Your paper will first be checked to make sure all elements meet the technical requirements. ASM staff will contact you if anything needs to be revised before copyediting and production can begin. Otherwise, you will be notified when your proofs are ready to be viewed.

Sincerely,
Marc Cook
Editor
mSystems

Reviewer #1 (Comments for the Author):

The resolution of Figure 2D & Figure 3 needs to be improved for better illustration.